# Summary of the Available Molecular Methods for Detection of SARS-CoV-2 during the Ongoing Pandemic

**DOI:** 10.3390/ijms22031298

**Published:** 2021-01-28

**Authors:** Fabio Arena, Simona Pollini, Gian Maria Rossolini, Maurizio Margaglione

**Affiliations:** 1Department of Clinical and Experimental Medicine, University of Foggia, 71122 Foggia, Italy; maurizio.margaglione@unifg.it; 2IRCCS Don Carlo Gnocchi Foundation, 50143 Florence, Italy; 3Department of Experimental and Clinical Medicine, University of Florence, 50134 Florence, Italy; simona.pollini@unifi.it (S.P.); gianmaria.rossolini@unifi.it (G.M.R.); 4Clinical Microbiology and Virology Unit, Florence Careggi University Hospital, 50134 Florence, Italy

**Keywords:** molecular methods, virus, pandemic, *Coronaviridae*, COVID-19, diagnosis, UK variant, B.1.1.7 variant, 501Y.V2 variant

## Abstract

Since early 2020, the COVID-19 pandemic has caused an excess in morbidity and mortality rates worldwide. Containment strategies rely firstly on rapid and sensitive laboratory diagnosis, with molecular detection of the viral genome in respiratory samples being the gold standard. The reliability of diagnostic protocols could be affected by SARS-CoV-2 genetic variability. In fact, mutations occurring during SARS-CoV-2 genomic evolution can involve the regions targeted by the diagnostic probes. Following a review of the literature and an in silico analysis of the most recently described virus variants (including the UK B 1.1.7 and the South Africa 501Y.V2 variants), we conclude that the described genetic variability should have minimal or no effect on the sensitivity of existing diagnostic protocols for SARS-CoV-2 genome detection. However, given the continuous emergence of new variants, the situation should be monitored in the future, and protocols including multiple targets should be preferred.

## 1. Background

In March 2020, the World Health Organization declared the severe respiratory disease caused by a new coronavirus (initially named “novel coronavirus 2019”: 2019nCoV), COVID-19 (Coronavirus Disease 2019), a global pandemic (https://www.who.int/dg/speeches/detail/who-director-general-s-opening-remarks-at-the-media-briefing-on-covid-19---11-march-2020). At the same time, the Coronaviridae Study Group (CSG) of the International Committee on Taxonomy of Viruses designated the virus responsible for COVID-19 as SARS-CoV-2 (severe acute respiratory syndrome coronavirus 2) [1].

Thenceforth, COVID-19 has continued expanding globally, causing almost 80 million infections and claiming more than 1.5 million lives from January 2020, as reported by the World Health Organization (situation updated as of 29 December 2020).

## 2. SARS-CoV-2 Genomic Features and Variability

SARS-CoV-2 belongs to the *Coronaviridae* family, subfamily *Orthocoronavirinae*. This subfamily is further divided into four genera, namely *Alphacoronavirus*, *Betacoronavirus*, *Gammacoronavirus*, and *Deltacoronavirus*. The majority of clinically relevant *Coronaviridae* belong to the *Alphacoronavirus* and *Betacoronavirus* [2]. The *Alphacoronavirus* and *Betacoronavirus* genera are currently divided into 12 and five subgenera, respectively, which are able to cause infections in a wide range of animal hosts (mainly bats but also cows, dogs, horses, pigs, and dromedaries). *Coronaviridae* infecting humans belong to the following subgenera: *Duvinacovirus* and *Setracovirus* for *Alphacoronavirus*, and *Embecovirus*, *Sarbecovirus*, and *Merbecovirus* for *Betacoronavirus* (https://www.ncbi.nlm.nih.gov/Taxonomy/Browser/wwwtax.cgi?mode=Undef&id=694002&lvl=3&p=has_linkout&p=blast_url&p=genome_blast&keep=1&srchmode=1&unlock). Subgenus *Merbecovirus* comprises the Middle East respiratory syndrome (MERS)-related coronaviruses. SARS-CoV-2, together with SARS-CoV (responsible for the 2002–03 SARS outbreak) are currently classified within the subgenus *Sarbecovirus*.

The *Betacoronaviruses*, like all other members of the *Coronaviridae* family, have relatively large RNA genomes of around 30 kb in size. The genomes have short untranslated regions (UTR) at both ends, with a 5′ methylated cap and a 3′ polyadenylated tail. Typically, *Coronaviridae* genomes contain 9–12 open reading frames (ORFs), six of which are conserved and follow the same order—the replicase/transcriptase polyproteins and the spike (S), envelope (E), membrane (M), and nucleocapsid (N) structural proteins. Replicase/transcriptase is organized in two overlapping ORFs, ORF1a (11–13 kb) and ORF1b (7–8 kb), which occupy nearly two-thirds of the genome. These ORFs are translated into two polyproteins that later cleave themselves to form several nonstructural proteins, most of which are involved in genome replication and translation [3].

The first whole-genome sequence of SARS-CoV-2 (strain Wuhan-HU-1) was deposited in the NCBI (National Center for Biotechnology Information) Genbank on 5 January 2020 [4]. Since then, the number of available genomes has increased dramatically during the pandemic, with thousands of SARS-CoV-2 whole-genome sequences available on the rapid data sharing service hosted by the Global Initiative on Sharing All Influenza Data (GISAID; https://www.epicov.org). Soon after the start of the pandemic, it seemed evident that SARS-CoV-2 is a recombinant virus between the bat coronavirus and a coronavirus of unknown origin [5].

The virus was first reported in the city of Wuhan, China [4,5,6], where an intermediate host, with a high probability, an animal sold at the seafood market in Wuhan, has likely facilitated the emergence of the virus in humans [7,8]. The early phases of dissemination of the virus outside China were linked to intercontinental travel originating to multiple introductions of different subclones in various geographic regions [9,10,11].

Even if members of the *Coronaviridae* family have the capacity of proofreading during genome replication, due to the presence of a non-structural exonuclease able to excise erroneous nucleotides inserted by the RNA polymerase [12], the SARS-CoV-2 global population has accumulated considerable genetic diversity at this stage of the COVID-19 pandemic [13]. Available data suggest that the SARS-CoV-2 genome accumulates variability at a rate of approximately 9.8 × 10^−4^ substitutions per site per year [13,14,15,16,17]. Mutations are generally rapidly purged from the viral population if highly deleterious. By contrast, neutral and advantageous mutations can reach higher frequencies. Some mutations may facilitate SARS-CoV-2 adaptation to the human host (decreasing virulence, increasing transmissibility, or escaping immune responses) and could emerge repeatedly and independently under natural selection. A series of small deletions across the whole-genome and single nucleotide polymorphisms (SNPs) occurring with high frequency have been identified and are summarized in Figure 1 [14].

Genomic variability allows the classification of several SARS-CoV-2 lineages. The two major classification efforts have been produced by GISAID (https://www.gisaid.org/references/statements-clarifications/clade-and-lineage-nomenclature-aids-in-genomic-epidemiology-of-active-hcov-19-viruses/) and Nextstrain (https://nextstrain.org/ncov) initiatives, respectively. Nextstrain assigns nomenclature through the designation of SARS-CoV-2 clades to label well-defined clades that reached geographic spread with significant frequency [19]. GISAID clade definitions are informed by the statistical distribution of genome distances in phylogenetic clusters, followed by the merging of smaller lineages into major clades based on shared marker variants [20]. The two systems produce largely overlapping phylogenetic trees [21].

The first isolates that appeared in Wuhan in December 2019 belonged to the L clade (GISAID classification). Its first variant, the S clade, appeared at the beginning of 2020, and from mid-January 2020, the V and G variants became prevalent. To date, clade G is the most widespread and has evolved in three subclades, namely, GR and GH, which appeared at the end of February 2020, and GV, which appeared later. Today, GR, GH, and GV are by far the most widespread in Europe. In North America, the most widespread is GH, while in South America GR seems prevalent. In Asia, where the Wuhan L strain initially appeared, the spread of G, GH, and GR is increasing. Globally, G, GH, and GR are constantly increasing, while S, L, and V strains are gradually disappearing [22,23].

More recently, a distinct phylogenetic cluster derived from the SARS-CoV-2 GR clade, named lineage B.1.1.7, has spread rapidly starting from early December 2020 in UK locations. The emergence of this variant is a cause of concern because it seems to be associated with increased transmissibility and an unusually large number of genetic changes, particularly in the spike protein (https://virological.org/t/preliminary-genomic-characterisation-of-an-emergent-sars-cov-2-lineage-in-the-uk-defined-by-a-novel-set-of-spike-mutations/563). However, some of the modifications occurring in the S protein of the B.1.1.7 lineage (e.g., the N501Y substitution and the deletion of six bases at positions 69 and 70, respectively, in the viral S gene) have been circulating globally for many months previously [21,24]. Another emerging lineage, named 501Y.V2, is characterized by some lineage-defining mutations in the spike protein, which has spread rapidly, becoming within weeks the dominant lineage in the Eastern Cape and Western Cape provinces (South Africa) [25].

Epidemiological investigations aiming at assessing new virus variants and their spread are useful in prioritizing relevant mutations and unraveling their potential impact on molecular diagnostics.

The information on genomic variability should be taken into account when a new diagnostic assay is released or when monitoring the reliability of already released methods. Ideally, diagnostics should target relatively invariant, strongly constrained regions of the SARS-CoV-2 genome, while multiple targets are preferred to increase detection sensitivity.

## 3. Diagnostic Tests for SARS-CoV-2 Infection

Diagnostic tests for SARS-CoV-2 infection belong to three categories, including (i) nucleic acid amplification tests (NAATs) detecting the presence of viral RNA by reverse transcription polymerase chain reaction (RT-PCR) or other amplification methods, (ii) tests detecting the presence of viral antigens, and (iii) tests detecting the presence of antibodies against SARS-CoV-2 antigens (Table 1).

NAATs detecting viral RNA in respiratory specimens remain the reference test for diagnosis of SARS-CoV-2 infection in patients with COVID-19-like symptoms and for patient triage and isolation in healthcare facilities, including long-term care facilities, outbreak investigations, and contact tracing activities. Testing at pre-determined time intervals can also be adopted as screening for infection in certain high-risk groups, such as healthcare workers and essential services workers, as part of surveillance programs. 

Soon after the emergence in China in January 2020, WHO announced several RT-PCR-based diagnostic schemes for SARS-CoV-2 based on the amplification of different viral targets (details available at https://www.who.int/docs/default-source/coronaviruse/whoinhouseassays.pdf), including some specific to SARS-CoV-2 (i.e., those targeting the viral RNA-dependent RNA polymerase-encoding RdRp gene and the viral nucleocapsid N gene) and one common to members of subgenus *Sarbecovirus* (i.e., the envelope E gene) (Table 2). The latter could also be used as a screening test, followed by the detection of SARS-CoV-2 specific targets [26,27,28]. The different viral targets were associated with different specificity and sensitivity, with the E gene being reported as the target with the highest sensitivity and the RdRp as the most specific [27].

More recently, additional assays based on the isothermal amplification of viral nucleic acids, also in combination with clustered regularly interspaced short palindromic repeat (CRISPR)-based detection methods, have been developed; these methods, which do not require thermal cycling, are generally more rapid than RT-PCR, declare good sensitivity and specificity, and are also considered suitable as point-of-care tests for the detection of SARS-CoV-2 [29,30,31].

Since the development of the first in-house diagnostic tests, several manufacturers have quickly developed commercial kits for molecular detection of SARS-CoV-2, based on existing diagnostic platforms. As a result, the number of commercial RT-PCR-based and isothermal nucleic acid amplification assays is at present considerably high and novel tests are continuously increasing the repertoire of available in vitro diagnostic assays (IVDs). Since the start of the pandemic, numerous tests have received the CE-IVD mark or the Food and Drug Administration (FDA) emergency use authorization (EUA) that is required to be placed in the market [32] (EUA assays are available at https://www.fda.gov/emergency-preparedness-and-response/mcm-legal-regulatory-and-policy-framework/emergency-use-authorization#covidinvitrodev). An online tool for existing SARS-CoV-2 assays performance comparison is available at https://www.finddx.org/covid-19/sarscov2-eval/. Furthermore, a meta-analysis focused on the diagnostic accuracy of point-of-care antigen and molecular-based methods for COVID-19 diagnosis has recently been published. The review underscored a high variability in the sensitivity of rapid tests across available studies (especially for antigen tests) [33]. The vast majority of these assays are RT-PCR schemes that require a separated viral RNA extraction step. Most of them target multiple viral genes (in most cases, the N and Orf1ab/RdRp genes), with a minority only being able to detect a single gene. The detection of multiple viral targets has the potential advantage of improving test sensitivity, particularly in case of low viral load in the initial specimen or RNA degradation during specimen handling, or in the event of viral genome mutations affecting one of the targeted regions [34]. In fact, test sensitivity is an important issue in the present scenario, where many assays that may differ in their capability of viral genome detection are proposed for laboratory diagnosis. Available commercial assays declare limits of detection in a quite large range (from less than 1 copy/PFU per mL to up to 1000 copies/PFU per mL) (https://www.fda.gov/emergency-preparedness-and-response/mcm-legal-regulatory-and-policy-framework/emergency-use-authorization#covidinvitrodev), possibly resulting in detection differences (particularly for low viral loads) among laboratories that use different diagnostic tests or in centers running side-by-side multiple assays on routine samples. However, the significance of extremely low viral loads remains to be ascertained, because in these cases, the SARS-CoV-2 quantity is apparently below the threshold at which replication-competent virus can be isolated by culture methods [35,36]. 

Although WHO diagnostic schemes have been deemed the gold standard at the beginning of the pandemic, they required specialized reagents, equipment, and personnel training. In the pandemic scenario, the possibility of rapidly scaling-up the number of tests and automation are crucial points in helping to face the ever-increasing number of required tests. Many commercially available tests can be automated by using robotic platforms able to separately extract viral RNA and prepare PCR assays, for high throughput batch processing of clinical specimens (Table 3). This approach may still require some significant expertise, dedicated equipment, and relatively long turn-around-times (TAT). At present, there are also a number of assays proposed as sample-to-result platforms (Table 3). Some of them (i.e., Panther/Aptima SARS-CoV-2 assay, Hologic Inc. Marlborough, MA; Cobas 6800/8800/cobas SARS-CoV-2, Roche, Basel, Swiss; and Alinity m System/Alinity m SARS-CoV-2 assay, Abbott Park, IL, USA) are high-throughput methods with a turnaround-time of approximately 2–3.5 h, while others can perform a smaller volume of tests with similar or reduced TAT (i.e., InGenius platform/SARS-CoV-2 ELITe MGB^®^ Kit and Simplexa COVID-19 Direct assay, ELITech, Pateaux, France), or are rapid single-test assays that give results in as fast as 13 min. (i.e., Abbott ID Now COVID-19 assay, Abbott Park, IL, USA) and up to 40–50 min. (e.g., Cepheid Xpert^®^ Xpress SARS-CoV-2 Cepheid, Sunnyvale, CA, USA; Bosch Vivalytic VRI test, Gerlingen-Schillerhöhe, Germany; and VitaPCR™ SARS-CoV-2 assay, Menarini, Florence, IT, USA) (Table 3). Some RT-PCR assays (e.g., BIOFIRE^®^ Respiratory Panel 2.1 bioMérieux, Marci l’etoile, France; and QIAstat-Dx Respiratory SARS-CoV-2 Panel, QUIAGEN, Hilden, Germany) are also developed as rapid syndromic panels; these tests are usually single-sample assays able to give results in up to one hour and may help healthcare providers to rapidly discriminate between common respiratory pathogens (e.g., flu and other viral pathogens) and SARS-CoV-2 in patients with COVID-19-like symptoms. Regarding the rapid assays, the availability of these tests appears to be of particular importance in managing suspect SARS-CoV-2-positive patients, mostly for fast patient triage and correct isolation procedures in the emergency departments. 

## 4. Influence of SARS-CoV-2 Genetic Variability on Molecular Diagnostic Protocols

Several studies have previously evaluated in silico the potential effect of mutations occurring in the target regions of published assays listed by WHO and other agencies (Table 4).

In June 2020, Khan and Cheung published an exhaustive evaluation of the sequence variability within the primer/probe target regions of the viral genome using more than 17,000 viral sequences from around the world [37]. Overall, the authors found a moderate mutation rate in the SARS-CoV-2 genome regions of interest. However, they reported a mismatch with all the viral sequences in the region of the Charité-ORF1b primer. Furthermore, they found a relatively high frequency of mutation in the region of the forward N gene primer released by the China CDC.

Three independent works confirmed, on a global and local scale, the overall high inclusivity of publicly available sets of primers and probes, with the exception of the forward N China CDC primer (occurrence of the so-called AAC variant) and set N3 of the US CDC, subsequently withdrawn [37,38,39,40,41].

In the largest bioinformatic project on this topic, Peñarrubia L. et al. analyzed nine different publicly available primers/probe sets with more than 30,000 genomes. The authors found a relatively high frequency of mutations in the regions of interest of various primers (approximately 34% of included genomes) and concluded that adopting multiple target approaches may mitigate the risk of loss of sensitivity [34] (Table 4). However, all these authors found only a small proportion of mutations involving the more problematic 3′ end of primers annealing regions; therefore, they concluded that, from a practical point of view, the impact of genetic variability on primers reliability should be minimal.

To expand the analysis to more recent sequences, we downloaded the selection of genomes present in the primary global analysis of the GISAID interactive database on December 9, 2020 (https://www.gisaid.org/epiflu-applications/phylodynamics/) and analyzed the variability in the WHO RT-PCR primers/probes regions of interest (Table 5). The GISAID system automatically subsamples 120 genomes per admin division (geographical area) per month to obtain a more geographically representative subset. We further customized the analysis, including only genomes obtained from human sources and sequences uploaded from 9 November 2020 to 9 December 2020.

A total of 1251 sequences were therefore included in the study and were available for further analysis (Appendix A). The majority of the sequences included in this study originated from Europe (27.6%), North America (26.6%), and Africa (21.1%), while there was only a relatively small number of sequences from Asia, Oceania, and South America (cumulatively 24.7%) (Figure 2). 

For the analysis of the presence of mismatches between PCR assays primers and probes and the selected SARS-CoV-2 genomes, we used a previously validated bioinformatic software pipeline [37]. Only the sequences variants with occurrence ≥1%, among included genomes were considered further and are shown in Table 5. The Wuhan-Hu-1/2019, 29,903 bp long genome (access. MN908947.3) was obtained from NCBI GenBank and used as a reference for site numbering and identification of primers and probes regions of interest.

Interestingly, most primer/probe binding regions showed no mutations or mutations/mismatches with a few sequences (one or two genomes, frequency <1%) representing probably extremely low prevalent variants or sequencing errors. Only in seven cases, the frequency of occurrence of single mutations reached 1%. The most relevant phenomenon was that of the circulation of the known AAC variant, causing a mismatch in the region of interest of the China CDC N forward primer. The variant was mainly found among recently sequenced genomes from Europe and North America, belonging to the GR and GH clades. Issues with this set of primers/probes were also linked to the presence of a mutation (G28975C) found in approximately 4% of genomes, in the reverse primer region. This mutation also was found among GH clade genomes from Europe.

Adjunctively, we expanded the analysis to more recent variants. UK B.1.1.7 variant has an unusually large number of genetic changes in the spike protein. Assays targeting the S-gene are not widely used for viral detection. Furthermore, relying on only one target for the detection of SARS-CoV-2 infection using RT-PCR is not recommended. However, the B.1.1.7 lineage shows a higher rate of molecular evolution, compared to other SARS-CoV-2 lineages, also outside the S gene. We screened a representative genome belonging to the B.1.1.7 variant (GISAID EPI_ISL_744131) for the presence of mismatches in regions of interest of the WHO RT-PCR primers/probes sets. Using the online NIH (National Institutes of Health) “Basic Local Alignment Search Tool,” we found a perfect match between primers/probes released by WHO (Table 2) and the genome sequence of the variant. As expected, the only mismatches found were in the first three positions of the China CDC N forward primer (“AAC variant”). The same analysis was conducted with the EPI_ISL_660190 genome, representative of the South Africa 501Y.V2 variant, and we found only two mismatches involving the central parts of China CDC N forward primer and the Japan National Institute of Infectious Diseases N reverse primer.

Less information is available on detection capabilities of commercial tests based on proprietary primers and probes. Many suppliers declare the ability of their test to in silico detect the viral variants that were circulating and included in freely available repositories (e.g., NCBI Genbank and GISAID) at the time of test approval; nevertheless, proprietary PCR primers sequences included in the assays are unavailable, preventing assessment by the users of the test’s ability to detect new viral variants.

A number of suppliers (e.g., Hologic, Roche, ELITEch, Diasorin, Seegene) have declared the ability of their tests to detect the B.1.1.7 SARS-CoV-2 variant.

## 5. Conclusions

Overall, considering previously published data, the bioinformatic analysis performed in this study and the information provided by companies producing commercial diagnostic systems, we can conclude that, currently, the known variability occurring in the SARS-CoV-2 population have minimal or no effect on the sensitivity of existing molecular systems for viral detection. Furthermore, the majority of mismatches observed were not near the 3′ end and should be tolerated.

The only exception that should be mentioned is the three nucleotide substitutions (GGG→AAC) that occur frequently in the three first positions of the China CDC N forward primer binding site. The so-called AAC variant is particularly frequent among the GR and GH recent clades.

Based on available sequencing data, the new UK B.1.1.7 and the South Africa 501Y.V2 variants should also be reliably recognized by the most widely used commercial and in-house protocols.

However, the continuous description of novel genomic variants represents an important diagnostic issue that needs to be monitored in the future, while a multiple target strategy is suggested to minimize the effect of SARS-CoV-2 genetic variability on assays sensitivity.

Given the increasing importance of genome sequencing data availability (to monitor the variability of the viral population), the development of rapid, inexpensive, and standardized laboratory/bioinformatic pipelines for SARS-CoV-2 genome sequencing is urgently needed.

## Figures and Tables

**Figure 1 ijms-22-01298-f001:**
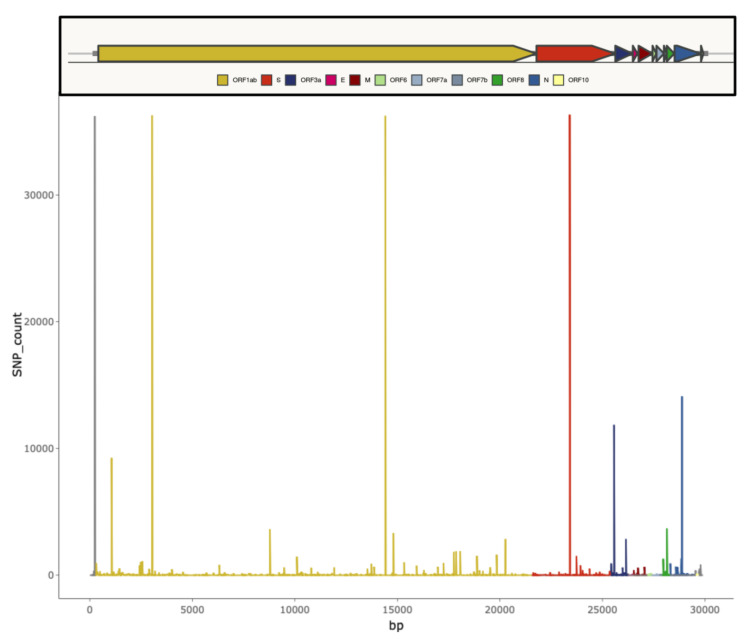
Graphical representation of SARS-CoV-2 genomic diversity derived from the alignment of 46,723 genomes obtained from different locations worldwide (from [18]). Vertical bars indicate the number of single nucleotide polymorphisms (SNPs) found for each genome position.

**Figure 2 ijms-22-01298-f002:**
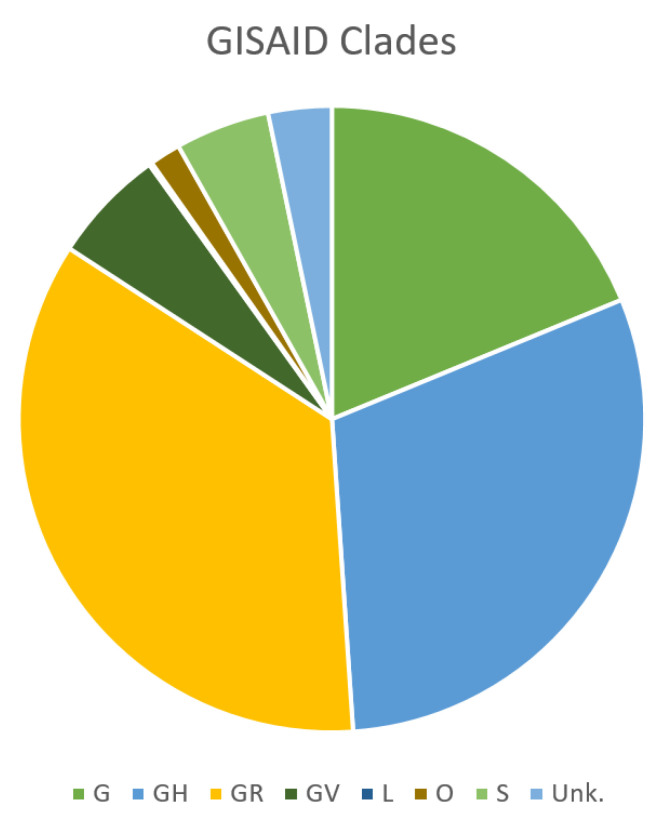
Subdivision in the Global Initiative on Sharing All Influenza Data (GISAID) clades of genomes included in this study.

**Table 1 ijms-22-01298-t001:** Summary of the main features of existing tests for COVID-19 diagnosis.

Assay Type	Principle of the Assay	Intended Use
Nucleic acid tests	detect the presence of viral RNA, generally by RT-PCR	decision making for clinical, infection control, or public health management (screening close contacts, outbreak investigations, or surveillance programs)
Antigen tests	detect the presence of a viral antigen, typically part of a surface protein, by lateral flow assays or chemiluminescence immunoassays	decision making for clinical, infection control, or public health management (screening close contacts, outbreak investigations, or surveillance programs)
Antibody tests	detect the presence of antibodies generated against SARS-CoV-2. The three most used assays are enzyme-linked immunosorbent assays, chemiluminescence assays, and lateral flow assays	sero-epidemiological surveys and studies; complement to the virus-detection tests

**Table 2 ijms-22-01298-t002:** Summary of primers/probes sets released by WHO for in-house reverse transcription polymerase chain reaction (RT-PCR) detection of SARS-CoV-2.

Source	Primer/Probe Name	Target Gene	Sequence	Lenght	Genomic Region *
China CDC, China	Forward (F)	ORF1ab	CCCTGTGGGTTTTACACTTAA	21	13,342–13,362
China CDC, China	Reverse (R)	ORF1ab	ACGATTGTGCATCAGCTGA	19	13,442–13,460
China CDC, China	Fluorescence probe (P)	ORF1ab	CCGTCTGCGGTATGTGGAAAGGTTATGG	28	13,377–13,404
China CDC, China	Forward (F)	N	GGGGAACTTCTCCTGCTAGAAT	22	28,881–28,902
China CDC, China	Reverse (R)	N	CAGACATTTTGCTCTCAAGCTG	22	28,958–28,979
China CDC, China	Fluorescence probe (P)	N	TTGCTGCTGCTTGACAGATT	20	28,934–28,953
Institut Pasteur, France	nCoV_IP2-12669Fw	RdRp	ATGAGCTTAGTCCTGTTG	18	12,690–12,707
Institut Pasteur, France	nCoV_IP2-12759Rv	RdRp	CTCCCTTTGTTGTGTTGT	18	12,780–12,797
Institut Pasteur, France	nCoV_IP2-12696bProbe(+)	RdRp	ATGTCTTGTGCTGCCGGTA	19	12,719–12,737
Institut Pasteur, France	nCoV_IP4-14059Fw	RdRp	GGTAACTGGTATGATTTCG	19	14,080–14,098
Institut Pasteur, France	nCoV_IP4-14146Rv	RdRp	CTGGTCAAGGTTAATATAGG	20	14,167–14,186
Institut Pasteur, France	nCoV_IP4-14084Probe(+)	RdRp	TCATACAAACCACGCCAGG	19	14,105–14,123
Institut Pasteur, France	E_Sarbeco_F1	E	ACAGGTACGTTAATAGTTAATAGCGT	26	26,269–26,294
Institut Pasteur, France	E_Sar beco_R2	E	ATATTGCAGCAGTACGCACACA	22	26,360–26,381
Institut Pasteur, France	E_Sarbeco_P1	E	ACACTAGCCATCCTTACTGCGCTTCG	26	26,332–26,357
US CDC, USA	2019-nCoV_N1-F	ORF9b	GACCCCAAAATCAGCGAAAT	20	28,287–28,306
US CDC, USA	2019-nCoV_N1-R	ORF9b	TCTGGTTACTGCCAGTTGAATCTG	24	28,335–28,358
US CDC, USA	2019-nCoV_N1-P	ORF9b	ACCCCGCATTACGTTTGGTGGACC	24	28,309–28,332
US CDC, USA	2019-nCoV_N2-F	ORF9b	TTACAAACATTGGCCGCAAA	20	29,164–29,183
US CDC, USA	2019-nCoV_N2-R	ORF9b	GCGCGACATTCCGAAGAA	18	29,213–29,230
US CDC, USA	2019-nCoV_N2-P	ORF9b	ACAATTTGCCCCCAGCGCTTCAG	23	29,188–29,210
US CDC, USA	2019-nCoV_N3-F	ORF9b	GGGAGCCTTGAATACACCAAAA	22	28,681–28,702
US CDC, USA	2019-nCoV_N3-R	ORF9b	TGTAGCACGATTGCAGCATTG	21	28,732–28,752
US CDC, USA	2019-nCoV_N3-P	ORF9b	ATCACATTGGCACCCGCAATCCTG	24	28,704–28,727
National Institute of Infectious Diseases, Japan	NIID_2019-nCOV_N_F2	N	AAATTTTGGGGACCAGGAAC	20	29,142–29,161
National Institute of Infectious Diseases, Japan	NIID_2019-nCOV_N_R2	N	TGGCAGCTGTGTAGGTCAAC	20	29,280–29,299
National Institute of Infectious Diseases, Japan	NIID_2019-nCOV_N_P2	N	ATGTCGCGCATTGGCATGGA	20	29,239–29,258
Charité, Germany	RdRP_SARSr-F2	RdRp	GTGAAATGGTCATGTGTGGCGG	22	15,431–15,452
Charité, Germany	RdRP_SARSr-R1	RdRp	CAAATGTTAAAAACACTATTAGCATA	26	15,505–15,530
Charité, Germany	RdRP_SARSr-P2	RdRp	CAGGTGGAACCTCATCAGGAGATGC	25	15,470–15,494
Charité, Germany	E_Sarbeco_F1	E	ACAGGTACGTTAATAGTTAATAGCGT	26	26,269–26,294
Charité, Germany	E_Sarbeco_R2	E	ATATTGCAGCAGTACGCACACA	22	26,360–26,381
Charité, Germany	E_Sarbeco_P1	E	ACACTAGCCATCCTTACTGCGCTTCG	26	26,332–26,357
HKU, HongKong SAR	HKU-ORF1b-nsp14F	ORF1b	TGGGGTTTTACAGGTAACCT	20	18,778–18,797
HKU, HongKong SAR	HKU-ORF1b-nsp14R	ORF1b	AACACGCTTAACAAAGCACTC	21	18,889–18,909
HKU, HongKong SAR	HKU-ORF1b-nsp141P	ORF1b	TAGTTGTGATGCAATCATGACTAG	24	18,849–18,872
HKU, HongKong SAR	HKU-NF	N	TAATCAGACAAGGAACTGATTA	22	29,145–29,166
HKU, HongKong SAR	HKU-NR	N	CGAAGGTGTGACTTCCATG	19	29,236–29,254
HKU, HongKong SAR	HKU-NP	N	GCAAATTGTGCAATTTGCGG	20	29,177–29,196
National Institute of Health, Thailand	WH-NICN-F	ORF9b	CGTTTGGTGGACCCTCAGAT	20	28,320–28,339
National Institute of Health, Thailand	WH-NICN-R	ORF9b	CCCCACTGCGTTCTCCATT	19	28,358–28,376
National Institute of Health, Thailand	WH-NICN-P	ORF9b	CAACTGGCAGTAACCA	16	28,341–28,356

* Site numbering uses Wuhan-Hu-1/2019 as reference (access.MN908947.3).

**Table 3 ijms-22-01298-t003:** The main CE-IVD and/or emergency use authorization (EUA)-labelled integrated extraction/amplification platforms and sample-to-result assays for the detection of SARS-CoV-2.

Assay	Manufacturer	Viral Genes	Assay/Equipment Type	Approx. Time-to-Result
Xpert^®^ Xpress SARS-CoV-2	Cepheid	N, E	RT-PCR/single test, sample-to-result	45 min.
Vivalytic analyzer/Vivalytic VRI test	BOSCH	Na ^a^	RT-PCR/single test, sample-to-result	39 min.
VitaPCR^TM^ platform/VitaPCR™ SARS-CoV-2 assay	Menarini	N	RT-PCR/single test, sample-to-result	20 min
GenMark ePlex instrument/ePlex^®^ SARS-CoV-2 Test	GenMark	N	RT-PCR/single test, sample-to-result	90 min.
ARIES^®^ SARS-CoV-2 Assay	Luminex Corporation	Orf1ab, N	RT-PCR/single test, sample-to-result	2 h
ID Now COVID-19	Abbott	RdRp	Isothermal amplification/single test, sample-to-result	13 min.
Simplexa COVID-19 Direct assay	DiaSorin	orf1ab, S	RT-PCR/batch testing, sample-to-result	80 min.
ELITech InGenius platform/SARS-CoV-2 ELITe MGB^®^ Kit	ELITech	RdRp, Orf8	RT-PCR/batch testing, sample-to-result	2 h 30 min.
Cobas 6800/8800/cobas SARS-CoV-2	Roche	orf1ab, E	RT-PCR/batch testing, sample-to-result	3 h 30 min.
Alinity m System/Alinity m SARS-CoV-2 assay	Abbott	RdRp, N	RT-PCR/batch testing, sample-to-result	2 h
NeoMoDx™ molecular system/NeuMoDx™ SARS-CoV-2 Assay	QIAGEN	Nsp2, N	RT-PCR/batch testing, sample-to-result	80 min.
BD MAX™ System/BD SARS-CoV-2 Reagents	Becton Dickinson	N	RT-PCR/batch testing, sample-to-result	3 h
Panther/Aptima SARS-CoV-2 assay	Hologic	orf1ab	Isothermal amplification/batch testing, sample-to-result	3 h 30 min.
Seegene NIMBUS/STARlet/Maelstrom 9600/Allplex™ SARS-CoV-2 Assay	Seegene	RdRp, N, S, E	RT-PCR/batch testing, integrated equipment for extraction and amplification ^b^	From 3 h 20 min. to 4 h 40 min.
KingFisher Flex Purification system/TaqPath™ COVID-19 RT-PCR Kit	Life Technologies Corporation	orf1ab, N, S	RT-PCR/batch testing, integrated equipment for extraction and amplification	na
BIOFIRE^®^ Respiratory Panel 2.1	Biomérieux	S, M	RT-PCR/syndromic panel, sample-to-result	45 min.
QIAstat-Dx Respiratory SARS-CoV-2 Panel	QIAGEN	RdRp, E	RT-PCR/syndromic panel, sample-to-result	60 min.

^a^ Not declared by the manufacturer. ^b^ Main assays using proprietary equipment for both extraction and amplification steps are reported.

**Table 4 ijms-22-01298-t004:** Summary of previously published papers analyzing the presence of mismatches between publicly available RT-PCR primers/probes and SARS-CoV-2 genomes.

No. of Genomes	No. of Primers/Probes Set Evaluated	Relevant Findings	Source	Period	Reference
17,027	27	100% of mutation frequency in the Charité-ORF1b and 18% in the forward primer of CN-CDC-N	GISAID	Genomes sequenced before 7 May 2020	[37]
992	10	mutations in the first 5′ three positions of the China CDC N forward primer, frequency 13%	GISAID	Genomes sequenced before 22 March 2020	[38]
2569	30	mutations in the first 5′ three positions of the China CDC N forward primer, frequency 14%	GISAID	Genomes sequenced before 7 April 2020	[39]
30	13	mutations in the China CDC N forward primer, frequency 16%	Locally sequenced genomes from Colombia	Period 6–24 March 2020	[40]
15,001	15	A single mismatch in the Charité group’s RdRP gene assay and the Japan NIID’s N gene assay; AAC variant at the 5′ end of the China CDC N forward primer, frequency 18.8%	GISAID	Genomes sequenced before 8 June 2020	[41]
33,819	9	AAC variant at the 5′ end of the China CDC N forward primer, frequency 24%	GISAID and GenBank	Genomes sequenced before June 2020	[34]

**Table 5 ijms-22-01298-t005:** Summary of mismatches between publicly available RT-PCR primers/probes and SARS-CoV-2 genomes analyzed in this study.

Source	Primer/Probe Name	Target Gene	Sequence	Lenght	Genomic Region *	Mutation	Frequency (%)	Clade Nextstrain	Clade GISAID	Country
China CDC, China	Forward (F)	N	GGGGAACTTCTCCTGCTAGAAT	22	28,881–28,902	G28881A	37.1	20A, 20B	G, GH, GR	Worldwide
						G28882A	36.9	20A, 20B	GH, GR	Worldwide
						G28883C	36.9	20A, 20B	GH, GR	Worldwide
						C28887T	2.9	19A, 20A, 20B, 20C	G, GH, GR, O	SriLanka
China CDC, China	Reverse (R)	N	CAGACATTTTGCTCTCAAGCTG	22	28,958–28,979	G28975C	4.6	20A	GH	Europe
US CDC	2019-nCoV_N3-P	ORF9b	ATCACATTGGCACCCGCAATCCTG	24	28,704–28,727	A28715T	2.0	20A, 20B	GH, GR	Japan
HKU, HongKong, SAR	HKU-NR	N	CGAAGGTGTGACTTCCATG	19	29,236–29,254	G29254A	1.0	20A, 20B, 20C	GH, GR	Latvia

* Site numbering uses Wuhan-Hu-1/2019 as reference (access.MN908947.3).

## Data Availability

Figure 1 was obtained from an open access and interactive web resource at https://macman123.shinyapps.io/ugi-scov2-alignment-screen/. SARS-CoV-2 genome data for the screening of mismatch with primers/probes sequences were downloaded from GISAID for SARS-CoV-2 online repository available at https://www.epicov.org. The online tool for comparison of SARS-CoV-2 diagnostic test performance is available at https://www.finddx.org/covid-19/sarscov2-eval/. The list of emergency use authorization (EUA) diagnostics is available at https://www.fda.gov/emergency-preparedness-and-response/mcm-legal-regulatory-and-policy-framework/emergency-use-authorization#covidinvitrodev. The genome sequence EPI_ISL_744131 was sequenced at Wales Specialist Virology Centre Sequencing lab and submitted to GISAID by Catherine Moore C. et al. The genome sequence EPI_ISL_660190 was sequenced at KRISP, KZN Research Innovation and Sequencing Platform, and submitted to GISAID by Giandhari J. et al. A.

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
