# Peer review of "Summary of the Available Molecular Methods for Detection of SARS-CoV-2 during the Ongoing Pandemic"

_ijms, 2021, doi:10.3390/ijms22031298_

Round 1
Reviewer 1 Report
It is a well written and comprehensive review by Arena and coworkers about summarize of the available molecular methods for detection of SARS-CoV-2 during the ongoing pandemic. The topic is current. I recommend for publication in IJMS after the following minor points are addressed.
- Figure 1 and 2 can’t be seen. The authors should upload the figures.
- For the readers, it is better that the authors could add one or two schematics to summarize some topic in the review, such as different diagnostic tests for SARS-CoV-2 infection.
- Please add some perspective for this topic in the last section.
Reviewer 2 Report
Reviesed paper "Summarize of the available molecular methods for detection of SARS-CoV-2 during the ongoing pandemic" is very interesting and very important in present strange time.
All Tables are perfect. In Review type papers very important is a meta-analysis, especially in presented topic of SARS-CoV-2. In this paper lack of meta-analysis. Paper needs major revision.
Round 2
Reviewer 2 Report
Accept in present form.